# High mRNA Expression Levels of Heat Shock Protein Family B Member 2 (*HSPB2*) Are Associated with Breast Cancer Patients’ Relapse and Poor Survival

**DOI:** 10.3390/ijms23179758

**Published:** 2022-08-28

**Authors:** Aimilia D. Sklirou, Despoina D. Gianniou, Paraskevi Karousi, Christina Cheimonidi, Georgia Papachristopoulou, Christos K. Kontos, Andreas Scorilas, Ioannis P. Trougakos

**Affiliations:** 1Department of Cell Biology and Biophysics, Faculty of Biology, National and Kapodistrian University of Athens, 15784 Athens, Greece; 2Department of Biochemistry and Molecular Biology, Faculty of Biology, National and Kapodistrian University of Athens, 15701 Athens, Greece; 3Department of Pathology, “Saint Savvas” Cancer Hospital of Athens, 11522 Athens, Greece

**Keywords:** biomarker, breast cancer, HSPB2, molecular chaperones, overall survival, prognosis

## Abstract

Small heat shock proteins (sHSPs) are ubiquitous ATP-independent chaperones that contribute to the maintenance of proteome integrity and functionality. Recent evidence suggests that sHSPs are ubiquitously expressed in numerous types of tumors and have been proposed to be implicated in oncogenesis and malignant progression. Heat shock protein family B member 2 (HSPB2) is a member of the sHSPs, which is found to be expressed, among others, in human breast cancer cell lines and constitutes an inhibitor of apical caspase activation in the extrinsic apoptotic pathway. In this study, we investigated the potential prognostic significance of *HSPB2* mRNA expression levels in breast cancer, which represents the most frequent malignancy in females and one of the three most common cancer types worldwide. To this end, malignant breast tumors along with paired non-cancerous breast tissue specimens were used. *HSPB2* expression levels were quantified in these two cohorts using a sensitive and accurate SYBR green-based quantitative real-time polymerase chain reaction (q-RT-PCR). Extensive biostatistical analyses were performed including Kaplan–Meier and Cox regression survival analyses for the assessment of the results. The significant downregulation of *HSPB2* gene expression was revealed in breast tumors compared to their adjacent non-cancerous breast tissues. Notably, high *HSPB2* mRNA expression predicts poor disease-free survival and overall survival of breast cancer patients. Multivariate Cox regression analysis revealed that *HSPB2* mRNA overexpression is a significant predictor of poor prognosis in breast cancer, independent of other clinicopathological factors. In conclusion, high *HSPB2* mRNA expression levels are associated with breast cancer patients’ relapse and poor survival.

## 1. Introduction

Breast cancer (BrCa) is the most frequent malignancy in females and one of the three most common cancers worldwide, followed by lung and colorectal cancer. In 2020, 2.3 million women were diagnosed with BrCa, and 685,000 deaths occurred globally [1,2]. Early-stage, non-metastatic BrCa, localized to the breast and local lymph nodes, is considered curable in ~70–80% of patients due to the improvements in multimodal therapy [3]. On the other hand, advanced BrCa with distant organ metastases (bones, lungs, liver, and brain), in addition to lymph nodes, is considered incurable by the currently available therapies, which aim to prolong the survival and maintain the patient’s quality of life [4,5].

BrCa is a complex heterogeneous disease classified into three major subtypes according to estrogen receptor (ER) or progesterone receptor (PR) expression and erb-b2 receptor tyrosine kinase 2 (ERBB2; formerly human epidermal growth factor 2, HER2) gene amplification. Notably, tumors expressing ER and/or PR are termed “hormone receptor-positive”, tumors expressing HER2 are called “HER2-positive”, and tumors lacking these three markers are considered “triple negative” [6,7]. Therapeutic approaches for BrCa differ according to the molecular subtype and generally include surgery, radiotherapy, chemotherapy, endocrine therapy, and/or targeted therapy [6,8].

Molecular chaperones, also known as heat shock proteins (HSPs) constitute a large family of molecular machines involved in the proper folding, unfolding, and assembly of polypeptides in order to maintain their structure and function [9,10]. In particular, HSPs curate the folding of nascent polypeptides into their native/functional configurations and prevent protein misfolding and aggregation [11,12,13]. They also target the misfolded or aggregated proteins for degradation, jointly with the protein quality control degradation machineries, i.e., the ubiquitin–proteasome system, and the autophagy–lysosome system, [14].

Several pieces of evidence suggest that HSPs are abnormally expressed in different types of cancer, including, among others, breast, colorectal, lung, prostate, pancreatic, bladder, and ovarian malignancies [15,16,17,18]. Recent advances in the field indicate that HSPs constitute potential biomarkers for cancer diagnosis and prognosis and are promising targets in cancer therapy [19,20].

Heat shock protein family B member 2 (HSPB2) is a member of the small HSPs (sHSPs), which consist of ubiquitous ATP-independent chaperones with low molecular weights [21,22,23]. HSPB2 is also named myotonic dystrophy protein kinase binding protein (MKBP), because it was found to bind to the myotonic dystrophy protein kinase (DMPK), thus increasing its activity and conferring thermal protection [24]. Although it is mainly expressed in cardiac and skeletal muscle [12,25], the *HSPB2* gene was also reported to be expressed in human breast cancer cell lines and constitutes an inhibitor of apical caspase activation in the extrinsic apoptotic pathway [26]. However, the role of HSPB2 in breast tumorigenesis remains largely obscure. In the current study, we examined *HSPB2* expression levels in BrCa tumors and matched adjacent normal tissue, and we also evaluated its potential association with patients’ relapse and overall survival.

## 2. Results

### 2.1. Downregulation of HSPB2 Gene Expression Levels during Oncogenic Transformation in Mammary Epithelial Cells

Firstly, we sought to examine whether the expression levels of the *HSPB2* gene are affected during oncogenic transformation. To this end, we used a genetically defined model of stepwise carcinogenesis, in which human primary mammary epithelial cells (HMEC) were immortalized by the expression of the human telomerase catalytic subunit (hTERT) (HME cells) followed by p53/pRb inactivation due to concomitant simian virus 40 large T antigen (LT) expression (HMLE cells); HMLE cells were then transformed by the co-expression of the oncogene H-RasV12, resulting in the generation of breast metastatic malignant cells (HMLER cells) [27]. Our analysis showed that *HSPB2* mRNA levels are significantly decreased in the HMLE and HMLER cell lines during oncogenesis as compared to normal HMEC cells (Figure 1); notably, this effect was maximized in HMLE cells suggesting a likely positive regulation of p53 and/or pRb in *HSPB2* gene expression.

### 2.2. Clinicopathological and Biological Characteristics of BrCa Patients and Used Samples

In the current study, two groups of samples were used, namely one group of 150 cancerous tissue specimens and another group of 16 paired non-cancerous tissue specimens from patients with primary BrCa. The cohort of patients consisted of 150 women, with a total median age of 60 years, ranging from 31 to 90 years, at the time of diagnosis. Moreover, concerning the histological grade and according to the World Health Organization (WHO) classification system, 8 patients were diagnosed with grade I (well-differentiated), 97 with grade II (moderately differentiated), and 45 with grade III malignant tumors (poorly differentiated). Furthermore, according to the TNM staging system, 43 malignant lesions were characterized as stage I (28.7%), 89 as stage II (59.3%), and 18 as stage III (12.0%). The clinical and biological traits of patients with BrCa included in the current study are shown in Table 1.

### 2.3. Reduced Expression Levels of HSPB2 mRNA in Breast Carcinoma Tissues as Compared with Paired Non-Cancerous Tissues

The initial comparison of *HSPB2* mRNA levels among 16 pairs of malignant breast tumors and their adjacent non-cancerous breast tissues revealed the downregulation of the *HSPB2* mRNA expression in the majority of malignant breast tumors (*p* < 0.001) (Figure 2). Specifically, the mean *HSPB2* mRNA expression levels were equal to 2215.7 RQU in tumor samples with a standard error of 265.8, while in non-cancerous samples the mean *HSPB2* mRNA expression was 11,234.5 RQU with a standard error of 3094.54 (Table 2). These findings corroborated our data in mammary epithelial cells suggesting that the *HSPB2* gene is most likely downregulated in malignant tumors.

To classify the *HSPB2* mRNA expression status in each tissue specimen as positive or negative, an optimal cut-off value was set, as described in the “Materials and Methods” section. We found that 75 (50%) samples were classified as *HSPB2* mRNA-positive, when the relative *HSPB2* mRNA expression was equal to or higher than 656.5 RQU and 75 (50%) samples as *HSPB2* mRNA-negative (relative *HSPB2* mRNA expression lower than 656.5 RQU). We then investigated the potential association of *HSPB2* mRNA expression status with patients’ clinicopathological parameters including molecular subtype, anatomic stage, mitotic rate, and HER2, ER, and PR status. We found no significant association between *HSPB2* gene expression levels and the mentioned clinicopathological parameters (not shown).

### 2.4. HSPB2 mRNA Overexpression Is a Reliable Predictor of Poor Prognosis in BrCa Patients, Independent of Other Clinicopathological Factors

The independent significance of *HSPB2* mRNA expression regarding patients’ relapse was revealed by the univariate Cox regression analysis (Table 3). Specifically, we found that BrCa patients with an *HSPB2*-positive expression status entailed a 2.29-fold higher risk of tumor recurrence as compared to *HSPB2*-negative ones (HR = 2.29, 95% CI = 1.19–4.42, *p* = 0.014). Kaplan–Meier curves revealed that *HSPB2*-positive BrCa patients have a significantly lower DFS compared to those who are *HSPB2*-negative (*p* = 0.011) (Figure 3A). The significance of the *HSPB2* mRNA expression status in the prognosis of patients’ DFS was also maintained in the multivariate Cox regression analysis (HR = 2.61, 95% CI = 1.34–5.08, *p* = 0.005). Consequently, high *HSPB2* mRNA expression levels represent an independent prognostic indicator for tumor recurrence in BrCa patients.

Then, we sought to examine the potential prognostic significance of *HSPB2* mRNA expression status for patients’ overall survival (OS). To this end, a univariable Cox regression analysis was performed and revealed that high *HSPB2* mRNA expression predicts poor OS for BrCa patients. Specifically, BrCa patients bearing *HSPB2*-positive tumors had a significantly shorter OS time interval than those with lower *HSPB2* mRNA levels (HR = 2.45, 95% CI = 1.23–4.96, *p* = 0.011). Furthermore, the molecular subtype and prognostic stage were found to be significant prognosticators of OS (Table 4).

The prognostic value of *HSPB2*-positive mRNA expression status in BrCa patients was also depicted by the Kaplan–Meier curves (*p* = 0.009) (Figure 3B). Additionally, multivariable Cox regression analysis revealed that *HSPB2* mRNA expression status remained a statistically significant factor of poor OS in BrCa, independent of molecular subtype and prognostic stage (HR = 2.69, 95% CI = 1.33–5.42, *p* = 0.006). Therefore, the *HSPB2*-positive mRNA expression status in BrCa could be considered a novel independent indicator of poor OS.

### 2.5. Prognostic Value of HSPB2 mRNA Expression in Patients with Breast Adenocarcinoma, Stratified According to Molecular Subtype, Anatomic Stage, or Prognostic Stage

Determination of tumor characteristics such as molecular subtype and tumor grade is crucial in the prognosis of patients diagnosed with BrCa [28]. Thus, patients were stratified according to these variables in order to evaluate the potential additional impact of *HSPB2* mRNA expression status. The stratification according to molecular subtype revealed that patients with triple-negative or HER2-positive tumors had significantly lower DFS (*p* < 0.001) and OS (*p* < 0.001) rates (Appendix A) compared to those with luminal A or luminal B tumors, regardless of the *HSPB2* expression status. Moreover, as depicted in Figure 4A, triple-negative patients positive for *HSPB2* mRNA expression showed an increased probability of poorer OS, as compared to those bearing *HSPB2*-negative tumors (*p* = 0.026).

Additionally, after stratification, according to the anatomic stage, patients with tumors of anatomic stage III were found to have a remarkably shorter DFS (*p* < 0.001) and OS (*p* < 0.001), compared to those with tumors of anatomic stage I or II, regardless of the *HSPB2* expression status (Appendix A). Furthermore, as illustrated in Kaplan–Meier curves, patients of anatomic stage II with *HSPB2*-positive tumors had remarkably lower DFS (Figure 5A; *p* = 0.006) and OS (Figure 4B; *p* = 0.011) rates, in comparison with those with *HSPB2*-negative tumors.

The stratification of patients according to prognostic stage revealed that patients of prognostic stage III had significantly shorter DFS (*p* < 0.001) and OS (*p* < 0.001) intervals, as compared to patients of stage I or II tumors, regardless of the *HSPB2* expression status (Appendix A). Furthermore, patients of prognostic stage II with *HSPB2*-positive tumors showed an increased probability of a poorer DFS (Figure 5B; *p* = 0.003) and OS (Figure 4C; *p* = 0.024), as compared to those with *HSPB2*-negative tumors.

## 3. Discussion

The network of molecular chaperones, also referred to as HSPs, plays a central role in the maintenance of proteome integrity and functionality (collectively referred to as proteostasis), and is of the utmost importance for cell homeodynamics and survival [29,30]. HSPs are involved in numerous cellular functions including protein degradation, stress tolerance, cell signaling, cell differentiation, and apoptosis [31,32,33]. Under stress conditions, cells induce the expression of HSPs, thus activating the heat shock response (HSR) [9,16,34]. Notably, sustained activation of HSR is often observed in cancer cells, as they experience increased levels of proteotoxic stress, which has been proposed as a stress hallmark of cancer [16,35,36].

BrCa is a highly heterogeneous disease characterized by not only various phenotypes and molecular subgroups, but also by different responses to treatment [37]. Depending on the tumor subtype, different kinds of therapies are applied, such as endocrine therapy for hormone receptor-positive disease [38] or anti-HER2 therapy in HER2-positive cases [39]. Despite significant medical achievements in its diagnosis, the biomarkers that are used in clinical practice today lack sensitivity and specificity [40,41]. Therefore, it is crucial to find novel non-invasive biomarkers that could ameliorate the estimation of patients’ recurrence and survival.

Over the last decades, growing evidence has shown that sHSPs expression is frequently deregulated in diverse cancer types and is proposed to profoundly impact malignant progression. In particular, sHSPs have been associated with several hallmarks of cancer, including tumorigenesis, cell growth, the evasion of apoptosis, immune surveillance, angiogenesis, metastasis, and chemoresistance [19,20,25,36,42,43,44].

HSPB2 expression is particularly elevated (despite ubiquitous expression) in cardiac and skeletal muscle [12,25] and was also found to be expressed in human BrCa cell lines [26]. Considering that BrCa is one of the most frequent malignancies, accounting for 11.7% of the total number of new cases diagnosed in 2020 [1,2] and that the role of HSPB2 in breast tumorigenesis or cancer progression has not yet been investigated, our study was focused on the potential prognostic significance of *HSPB2* mRNA expression levels in BrCa patients.

Despite the fact that many sHSPs are overexpressed in a wide range of solid tumors [45], we observed that *HSPB2* gene transcription is predominantly downregulated in malignant breast tumors compared to their adjacent non-cancerous breast tissues. In line with our findings, a recent comprehensive transcriptomic study of the HSP gene family in BrCa patients from both The Cancer Genome Atlas (TCGA) and the Molecular Taxonomy of Breast Cancer International Consortium (METABRIC) cohorts revealed that *HSPB2* gene expression levels are profoundly downregulated in all BrCa molecular subtypes as compared to normal breast tissues [46]. Salhia et al. have also demonstrated the underexpression of *HSPB2* in BrCa samples, due to *HSPB2* gene deletion [47]. Other studies support the downregulation of *HSPB2* in cancer, as well; specifically, *HSPB2* mRNA was lower in esophageal squamous cell carcinoma cell lines, due to hypermethylation of the promoter of the *HSPB2* gene [48], while *HSPB2* mRNA was also found to be barely expressed in pancreatic cancer [49]. Since all the aforementioned studies note the lower mRNA expression of *HSPB2*, whereas most studies stating the overexpression of HSPs have examined their protein levels, a possible explanation is that HSPB2 protein expression levels do not perfectly correlate with *HSPB2* mRNA levels. Indeed, as shown by transcriptomics and proteomics data of *HSPB2* expression in various BrCa cell lines, deposited to the Expression Atlas database [50], the lower mRNA levels are not reflected by lower protein expression levels of this molecule.

Multivariate Cox regression analysis revealed that *HSPB2* mRNA overexpression is a significant predictor of poor prognosis in BrCa, independent of other clinicopathological factors, including molecular subtype and prognostic stage. Additionally, Kaplan–Meier survival curves revealed that patients with *HSPB2*-positive tumors were more likely to have poor outcomes, such as relapse and death. Consistently, the elevated levels of diverse HSPs expression in specific malignancies have been associated with a poor prognosis and an increased resistance to therapies [44,45,51,52,53].

Heat shock factor 1 (HSF1) constitutes the most robust regulator of HSPs expression to maintain (among others) proteome stability [54]. A growing number of studies support that HSF1 is implicated in the initiation, promotion, and progression of cancer and is widely exploited as a potential therapeutic target in a broad spectrum of malignancies [16,55]. In line with our findings, increased expression levels of HSF1 were found to be associated with a poor prognosis in BrCa patients [16,56,57].

Furthermore, although the role of HSPB2 has not been investigated so widely as the role of other HSPs, it was found to confer resistance to apoptosis in human BrCa cell lines, as it inhibited the apical caspase activation in the extrinsic apoptotic pathway [26], as well as to inhibit pancreatic cancer cell proliferation via activating targets of the TP53 signaling pathway, such as the *RPRM*, *ADGRB1*, and *STEAP3* genes [49]. In addition, it has recently been hypothesized that the inhibition of HSPB2 by miR-17-5p promotes cell proliferation, migration, and invasion of colon cancer cell lines [58]. The constitutive activation of HSPs in various malignancies is proposed to confer a survival advantage to cancer cells [59,60]. Therefore, HSPs could exert oncogenic functions and mediate the “non-oncogene addiction” of cancer cells being crucial for tumor development and survival [36,59,60,61,62].

The major implication of HSPs in cell transformation and tumor progression largely supports the notion that HSPs-targeting drugs could constitute a promising approach in cancer therapy [52,60,63,64]. Notably, HER2 is a protein client of the heat shock protein 90 (HSP90) and thus, HSP90 inhibitors have already been used in clinical trials in HER2 positive BrCa patients [65,66]. Additionally, trastuzumab (anti-HER2 monoclonal antibody) resistance, which is associated with a poorer prognosis, was attenuated in HER2-positive BrCa through HSP90 inhibition [58,67] and could serve as an adverse, independent prognostic biomarker for this malignancy. Nevertheless, our study is characterized by some limitations that need to be addressed. Firstly, our cohort size is of a medium size, and the number of non-cancerous breast tissue specimens is rather small. In addition, the patients’ cohort was not equivalently stratified in the defined subgroups, which could diminish the obtained findings. Future studies should be conducted to further evaluate the role of HSPB2 in BrCa prognosis.

Molecular biomarkers represent biological molecules used to infer disease risk, diagnosis, prognosis, and therapeutic response [68]. It is evident that the identification of accurate and precise non-invasive molecular biomarkers constitutes the main tool for paving the way toward “individualized biomarker-driven cancer therapy” or otherwise “precision medicine”, so that optimal treatment decisions can be made [69,70]. Consequently, *HSPB2* mRNA expression status could be combined with other well-validated clinical biomarkers, which could tailor the therapeutic options aiming to improve the outcome of cancer therapy and a patient’s overall survival, while minimizing the associated risk.

## 4. Materials and Methods

### 4.1. Cell Lines and Cell Culture Conditions

Human mammary epithelial cells HMEC were obtained from Lonza Group AG (Basel, Switzerland), while the human mammary adenocarcinoma cell line MCF-7 was obtained from the American Tissue Culture Collection (Manassas, VA, USA). The HMEC-derived cell lines HME, HMLE, and HMLER were a kind gift by Prof. Robert A. Weinberg (Massachusetts Institute of Technology, Cambridge, MA, USA). HMEC, HME, HMLE, and HMLER cells were grown in MEGM™ Mammary Epithelial Cell Growth Medium BulletKit™ (Lonza Group AG, #CC-3150). MCF-7 cells were cultured in DMEM (Thermo Fisher Scientific Inc., Waltham, MA, USA) supplemented with 2 g/L glucose, 10% FBS, and 1% penicillin/streptomycin. All cell lines were maintained in a humidified incubator at 5% CO_2_, 95% humidity, and 37 °C.

### 4.2. Patients and Tissue Collection

One hundred and fifty (150) BrCa samples and 16 paired, non-cancerous tissue samples were collected from patients with primary BrCa, subjected to surgery at the “Saint Savvas” Cancer Hospital of Athens, Athens, Greece. The biological and clinicopathological data collected included the age of patients, the dimensions of the resected tumor, the infiltration of regional lymph nodes, the presence of distant metastasis, the histological and intrinsic (molecular) subtypes, as well as the histological grade of the tumor, the expression status of PR, ER, HER2, and the mitotic rate based on the Ki-67 index. All tumors were independently characterized by two pathologists. The anatomic (TNM) and prognostic stages were determined, based on these data. All breast tissue specimens were stored in liquid nitrogen immediately after surgery. Survival data were available for 145 out of 150 patients, and the median follow-up time was 97 months.

The study was conducted in compliance with the 1964 Declaration of Helsinki and its later amendments and was approved by the institutional Ethics Committee of the “Saint Savvas” Cancer Hospital of Athens, Athens, Greece. Written informed consent was obtained from all patients.

### 4.3. Total RNA Extraction and First-Strand cDNA Synthesis

Total RNA was extracted from each sample using the TRI reagent^®^ (Molecular Research Center, Inc., Cincinnati, OH, USA), following the manufacturer’s protocol instructions. Spectrophotometric evaluation of the concentration and purity of the isolated RNA samples was conducted using a BioSpec-nano Micro-volume UV–Vis Spectrophotometer (Shimadzu, Kyoto, Japan). Two micrograms μg of total RNA were subjected to reverse transcription, using M-MLV reverse transcriptase (Life Technologies Ltd., Carlsbad, CA, USA) and oligo-dT 5 μM, according to the manufacturer’s instructions, to obtain first-strand cDNA.

### 4.4. Quantitative Real-Time PCR (qPCR)

A real-time qPCR assay was developed and carried out, using the SYBR Green Chemistry, in an ABI 7500 Real-Time PCR System (Applied Biosystems, Foster City, CA, USA). Specific primers were designed for *HSPB2*; moreover, specific primer pairs were designed for housekeeping genes encoding glyceraldehyde-3-phosphate dehydrogenase (*GAPDH*) and hydroxymethylbilane synthase (*HMBS*), which were chosen among others as reference genes, according to supporting literature data [71] and our preliminary results in a random sample of our cohort of BrCa specimens. The sequences of *HSPB2* primers were 5′-CCGAGTACGAATTTGCCAACC-3′ and 5′-AGGCCGGACATAGTAGCCAT-3′. Those of *GAPDH* were 5′-GTCAAGGCTGAGAACGGGAA-3′ and 5′-TCGCCCCACTTGATTTTGGA-3′, and those of *HMBS* were 5′-AAGAGACCATGCAGGCTACCA-3′ and 5′-ACAAGTTGGCCAGGCTGATG-3′.

The qPCR mixture contained KAPA SYBR FAST qPCR Master Mix Universal, supplemented with ROX as a passive reference dye, forward and reverse primers at a final concentration of 200 nM, 2.5 μL DEPC-treated H_2_O, and 0.5 μL of 10-fold diluted cDNA. We optimized the qPCR assays by standardizing the primer concentration and the thermal protocol, so as to observe a unique melting curve for each assay. The uniqueness and specificity of each amplicon were also assessed by agarose gel electrophoresis. The cDNA derived from the MCF-7 cells was used as a calibrator to render “−∆Ct” values calculated in distinct qPCR runs comparable. The comparative Ct method (2^−∆∆Ct^) was applied for relative quantification [72,73]. The normalized *HSPB2* expression of each sample was defined as the ratio of *HSPB2* molecules to *GAPDH* and *HMBS* molecules, divided by the same ratio that was calculated for the calibrator (MCF-7 cells) and was determined in relative quantification units (RQU).

### 4.5. Biostatistical Analysis

The distribution in the patients’ cohort was not normal; thus, non-parametric tests were used. Firstly, descriptive statistics were performed. The Wilcoxon signed-rank test was carried out to assess the difference in *HSPB2* mRNA expression between paired samples. Next, the BrCa patients were categorized into two distinct groups, namely *HSPB2*-positive and *HSPB2*-negative patients; this categorization was done based on the median value of *HSPB2* mRNA expression (656.5 RQU). More specifically, patients with an *HSPB2* mRNA expression value higher than 656.5 RQU were characterized as *HSPB2*-positive, whereas those with an *HSPB2* mRNA expression value lower than 656.5 RQU were characterized as *HSPB2*-negative. Potential associations between the expression of *HSPB2* mRNA and other categorical clinicopathological variables were investigated, using a two-tailed χ^2^-test.

Kaplan–Meier survival analysis was then applied, with regard to disease-free survival (DFS) and overall survival (OS). Differences between Kaplan–Meier curves were assessed using the log-rank (Mantel–Cox) test. Bootstrap (1000 random samples) univariate and multivariable Cox regression analyses were carried out, concerning also DFS and OS; further, the bias-corrected and accelerated (BCa) 95% confidence interval (CI) of each hazard ratio (HR) was estimated. Next, Kaplan–Meier survival analysis was performed in subgroups of the cohort, in which patients were stratified according to specific clinicopathological characteristics. Each outcome was considered statistically significant if the *p*-value < 0.050.

## 5. Conclusions

Our findings suggest that the *HSPB2* molecular chaperone gene transcription is predominantly downregulated in BrCa, whereas increased *HSPB2* gene expression levels are associated with patients’ relapse and poor survival. Consequently, *HSPB2* mRNA expression status could be potentially used to assess the clinical prognosis of patients with this frequent malignancy, independent of other clinicopathological factors.

## Figures and Tables

**Figure 1 ijms-23-09758-f001:**
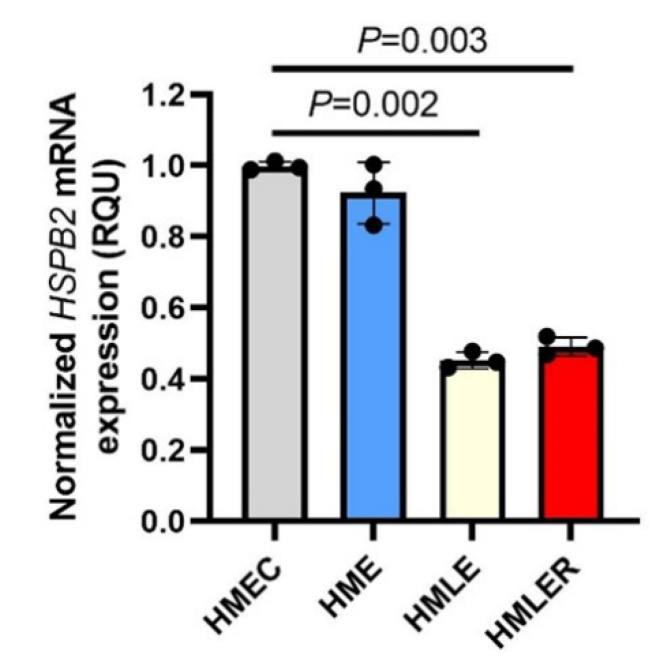
Downregulation of the *HSPB2* mRNA levels during the oncogenic transformation of mammary epithelial cells. Relative *HSPB2* mRNA expression levels in the genetically defined model of stepwise carcinogenesis consisting of HMEC (normal), HME, HMLE, and HMLER cells. Control HMEC cells were set to 1. *GAPDH* gene expression was used as a reference for RNA input. *p*-values were calculated using an unpaired *t*-test.

**Figure 2 ijms-23-09758-f002:**
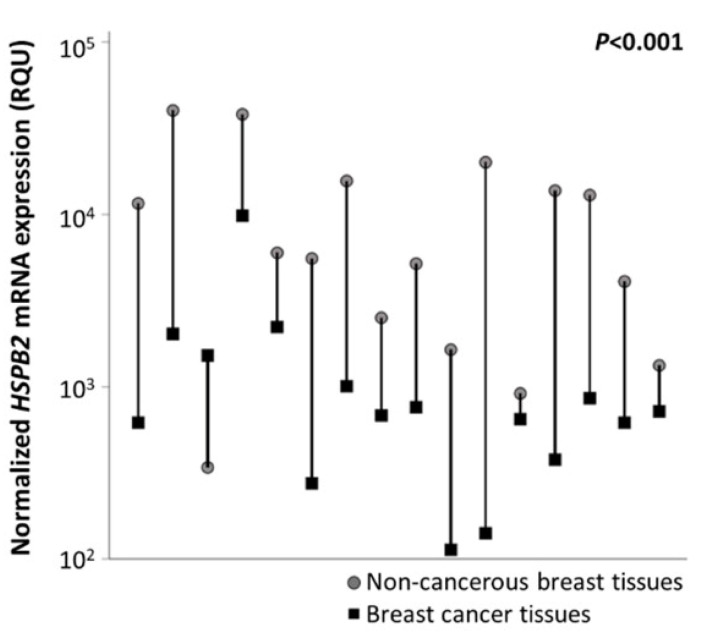
Graphical illustration of *HSPB2* mRNA expression levels in cancerous vs. non-cancerous tissues, after comparing 16 pairs of tissue specimens. The *HSPB2* mRNA expression levels were downregulated, as compared to normal adjacent tissues, in almost all tumors. *p*-value was calculated by the Wilcoxon signed-rank test.

**Figure 3 ijms-23-09758-f003:**
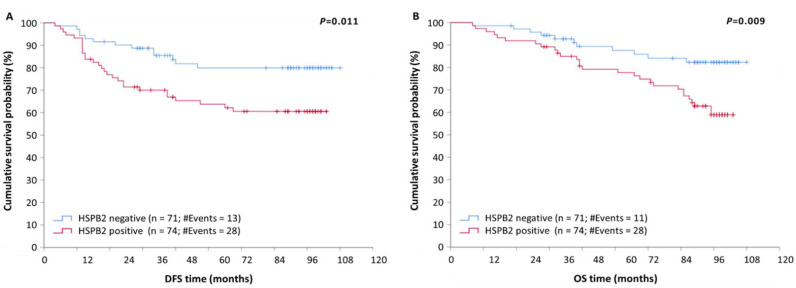
Kaplan–Meier survival curves for disease-free survival (DFS) and overall survival (OS) of BrCa patients. (**A**) Patients with tumors being positive for *HSPB2* mRNA expression had shorter DFS time intervals than patients with *HSPB2*-negative tumors (*p* = 0.011). (**B**) Patients with *HSPB2*-positive tumors had shorter OS than those with *HSPB2*-negative tumors (*p* < 0.009). *p*-value was calculated using the log-rank test.

**Figure 4 ijms-23-09758-f004:**
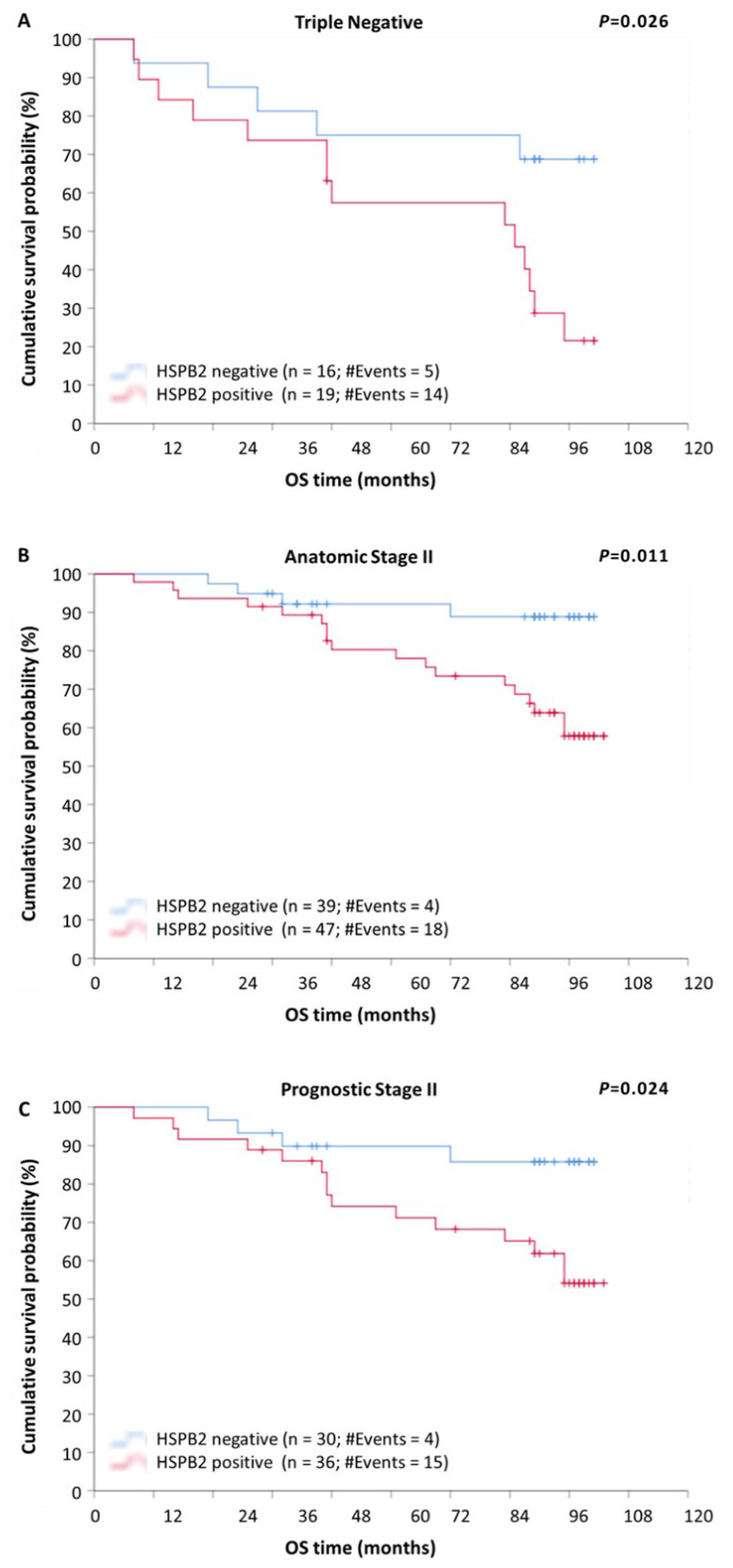
Stratified Kaplan–Meier survival curves for overall survival (OS) of BrCa patients, according to molecular subtype, anatomic, and prognostic stage. (**A**) Patients with triple-negative tumors being positive for *HSPB2* mRNA expression had shorter OS time intervals than patients with *HSPB2*-negative tumors. (**B**) Patients with *HSPB2*-positive tumors of anatomic stage II had poorer OS than those with *HSPB2*-negative tumors. (**C**) Patients with *HSPB2*-positive tumors of prognostic stage II showed an increased probability of poorer OS compared to those with *HSPB2*-negative tumors. *p*-value was calculated using the log-rank test.

**Figure 5 ijms-23-09758-f005:**
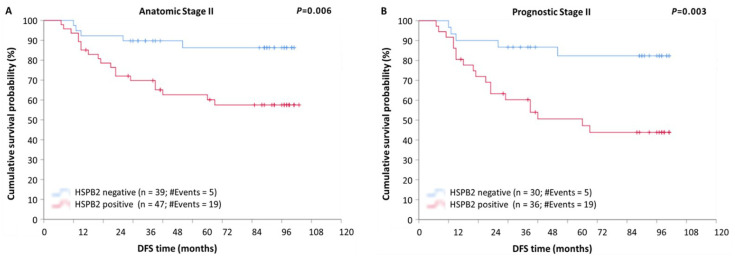
Stratified Kaplan–Meier survival curves for disease-free survival (DFS) of BrCa patients, according to anatomic and prognostic stage. (**A**) Patients with *HSPB2*-positive tumors of anatomic stage II had significantly poorer DFS than those with *HSPB2*-negative tumors. (**B**) Patients with *HSPB2*-positive tumors of prognostic stage II had shorter DFS time intervals than those with *HSPB2*-negative tumors. *p*-value was calculated using the log-rank test.

**Table 1 ijms-23-09758-t001:** Clinicopathological features of BrCa patients.

	Number of Patients (%)
**Patients**	150
***HSPB2* mRNA Expression Status**	
Negative	75 (50.0%)
Positive	75 (50.0%)
**Age (y)**	Median: 60; Range: 31–90
**Tumor Size (cm^2^)**	Median: 2.35; Range: (0.60–8.50)
**Cancer Type**	
Invasive ductal carcinoma	121 (80.7%)
Lobular carcinoma	16 (10.7%)
Other carcinomas	13 (8.6%)
**Molecular Subtype**	
Luminal A	59 (39.3%)
Luminal B	40 (26.7%)
Triple-negative	36 (24.0%)
HER2-enriched	15 (10.0%)
**Histological Grade**	
I	8 (5.3%)
II	97 (64.7%)
III	45 (30.0%)
**HER2 Status**	
Negative	115 (79.3%)
Positive	30 (20.7%)
Unknown	5
**ER Status**	
Negative	56 (37.3%)
Positive	94 (62.7%)
**PR Status**	
Negative	82 (54.7%)
Positive	68 (45.3%)
**Ki-67 Index**	
Low (≤14%)	92 (63.9%)
High (>14%)	52 (36.1%)
Unknown	6
**Anatomic Stage**	
I	43 (28.7%)
II	89 (59.3%)
III	18 (12.0%)
**Prognostic Stage**	
I	60 (40.0%)
II	68 (45.3%)
III	22 (14.7%)

Abbreviations: ER, estrogen receptors; PR, progesterone receptors.

**Table 2 ijms-23-09758-t002:** Distributions of *HSPB2* mRNA expression levels in cancerous and non-cancerous tissue samples.

Variable	Mean ± SE	Range	Percentiles		
			25th	50th (Median)	75th
*HSPB2* mRNA Expression (RQU)					
in breast cancer tissues (*n* = 150)	2215.7 ± 265.8	29.0–19,101.0	315.5	656.5	2637.8
in non-cancerous tissues (*n* = 16)	11,234.5 ± 3094.54	340.0–40,093.0	1864.0	5778.5	15,170.3

Abbreviations: RQU, relative quantification units; SE, standard error.

**Table 3 ijms-23-09758-t003:** *HSPB2* mRNA expression status and disease-free survival (DFS) of BrCa patients.

	Univariate Analysis (*n* = 145)					Multivariable Analysis ^1^ (*n* = 145)				
Covariate	HR ^2^	95% CI ^3^	*p* Value ^4^	BCa Bootstrap 95% CI ^3^	Bootstrap *p* Value ^4^	HR ^2^	95% CI ^3^	*p* Value ^4^	BCa Bootstrap 95% CI ^3^	Bootstrap *p* Value ^4^
*HSPB2* mRNA expression status										
Negative (*n* = 71)	1.00					1.00				
Positive (*n* = 74)	2.29	1.19–4.42	*0.014*	1.17–4.82	*0.014*	2.61	1.34–5.08	*0.005*	1.26–6.61	*0.006*
Molecular subtype					*<0.001*			0.071		
Luminal A (*n* = 56)	1.00					1.00				
Luminal B (*n* = 39)	1.02	0.36–2.88	0.97	0.31–3.11	0.97	1.27	0.45–3.60	0.65	0.24–4.65	0.66
Triple-negative (*n* = 35)	3.63	1.60–8.21	*0.002*	1.45–10.88	*0.003*	2.04	0.83–5.02	0.12	0.78–7.18	0.094
HER2-enriched (*n* = 15)	6.04	2.44–14.96	<*0.001*	2.40–19.65	<*0.001*	3.59	1.35–9.20	*0.010*	1.42–11.78	*0.005*
Prognostic stage					*<0.001*			*0.013*		
I (*n* = 58)	1.00					1.00				
II (*n* = 66)	5.00	1.90–13.10	*0.001*	1.78–4.89 × 10^4^	*0.002*	3.72	1.33–10.38	*0.012*	0.99–1.10 × 10^5^	*0.010*
III (*n* = 21)	10.06	3.53–28.63	<*0.001*	2.87–87.66	<*0.001*	6.22	1.83–21.12	*0.003*	1.4–1.45 × 10^6^	*0.005*

^1^ Multivariable models regarding DFS were adjusted for molecular subtype and prognostic stage. ^2^ Hazard ratio, estimated from proportional hazard Cox regression models. ^3^ Confidence interval of the estimated HR. ^4^ Statistically significant *p* values are shown in italics.

**Table 4 ijms-23-09758-t004:** *HSPB2* mRNA expression status and overall survival (OS) of BrCa patients.

	Univariate Analysis (*n* = 145)					Multivariable Analysis ^1^ (*n* = 145)				
Covariate	HR ^2^	95% CI ^3^	*p* Value ^4^	BCa Bootstrap 95% CI ^3^	Bootstrap *p* Value ^4^	HR ^2^	95% CI ^3^	*p* Value ^4^	BCa Bootstrap 95% CI ^3^	Bootstrap *p* Value ^4^
*HSPB2* mRNA expression status										
Negative (*n* = 71)	1.00					1.00				
Positive (*n* = 74)	2.45	1.23–4.96	*0.011*	1.21–5.79	*0.015*	2.69	1.33–5.42	*0.006*	1.23–7.98	*0.007*
Molecular subtype			<*0.001*					0.085		
Luminal A (*n* = 56)	1.00					1.00				
Luminal B (*n* = 39)	1.16	0.40–3.35	0.78	3.66–38.89	0.79	1.27	0.44–3.66	0.66	0.30–4.40	0.67
Triple-negative (*n* = 35)	4.66	2.04–10.66	<*0.001*	15.17–3.89 × 10^6^	<*0.001*	3.22	1.26–8.23	*0.014*	1.26–10.23	*0.008*
HER2-enriched (*n* = 15)	3.54	1.23–10.22	*0.019*	12.50–2.69 × 10^5^	*0.011*	2.45	0.79–7.59	0.12	0.47–10.17	0.14
Prognostic stage								0.16		
I (*n* = 58)	1.00					1.00				
II (*n* = 66)	2.26	0.99–5.16	0.053	0.96–6.20	*0.046*	1.62	0.67–3.92	0.29	0.64–4.88	0.23
III (*n* = 21)	6.00	2.44–14.71	<*0.001*	2.29–17.55	<*0.001*	2.86	0.96–8.53	0.059	0.82–15.54	*0.049*

^1^ Multivariable models regarding OS were adjusted for molecular subtype and prognostic stage. ^2^ Hazard ratio, estimated from proportional hazard Cox regression models. ^3^ Confidence interval of the estimated HR. ^4^ Statistically significant *p* values are shown in italics.

## Data Availability

The data presented in this study are available on reasonable request from the corresponding authors.

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
