# Peer review of "High mRNA Expression Levels of Heat Shock Protein Family B Member 2 (HSPB2) Are Associated with Breast Cancer Patients’ Relapse and Poor Survival"

_ijms, 2022, doi:10.3390/ijms23179758_

Round 1
Reviewer 1 Report
The authors describe some interesting and unusual phenomenon: decrease HSPB2 mRNA in malignant breast tissue comparing to normal tissues, and, at the same time, increase in this mRNA in patients with relapse and poor survival. So far, in most cases levels of chaperones (e.g., Hsp70, HSPB1) were found to increase both during transformation and cancer progression. Thus, if HSPB2 plays anti-apoptotic role as the authors mention, it is completely unclear why its levels drops in cancer where anti-apoptotic protection is indispensable feature and one may expect rather increase than decrease in its levels. The major concern of this work is that the authors studied only expression of HSPB2 mRNA rather than protein which was done in most previous studies, and it is well-known that protein expression (which is more relevant) may not correlate with mRNA expression. Therefore, it would be much more convincing if the authors find some data in databases or elsewhere regarding correlation between expression of HSPB2 mRNA and protein.
Another point is that no description of how cutoff value cutoff between 2 groups (with high and low HSPB2) was set (Page 3, line 146)
Also, at least 2 refs are absent:
1. Chang et al. Promoter methylation of heat shock protein B2 in human esophageal squamous cell carcinoma. Int J Oncol 38: 1129 (2011), where decrease in HspB2 mRNA was found in malignant tissue.
2. Salhia et al. Integrated Genomic and Epigenomic Analysis of Breast Cancer Brain Metastasis. PLOS Volume 9 | Issue 1 | e85448 (2014), where underexpression/deletion of HspB2 gene was observed in Breast Cancer Brain Metastasis.
The authors describe some interesting and unusual phenomenon: decrease HSPB2 mRNA in malignant breast tissue comparing to normal tissues, and, at the same time, increase in this mRNA in patients with relapse and poor survival. So far, in most cases levels of chaperones (e.g., Hsp70, HSPB1) were found to increase both during transformation and cancer progression. Thus, if HSPB2 plays anti-apoptotic role as the authors mention, it is completely unclear why its levels drops in cancer where anti-apoptotic protection is indispensable feature and one may expect rather increase than decrease in its levels. The major concern of this work is that the authors studied only expression of HSPB2 mRNA rather than protein which was done in most previous studies, and it is well-known that protein expression (which is more relevant) may not correlate with mRNA expression. Therefore, it would be much more convincing if the authors find some data in databases or elsewhere regarding correlation between expression of HSPB2 mRNA and protein.
Another point is that no description of how cutoff value cutoff between 2 groups (with high and low HSPB2) was set (Page 3, line 146)
Also, at least 2 refs are absent:
1. Chang et al. Promoter methylation of heat shock protein B2 in human esophageal squamous cell carcinoma. Int J Oncol 38: 1129 (2011), where decrease in HspB2 mRNA was found in malignant tissue.
2. Salhia et al. Integrated Genomic and Epigenomic Analysis of Breast Cancer Brain Metastasis. PLOS Volume 9 | Issue 1 | e85448 (2014), where underexpression/deletion of HspB2 gene was observed in Breast Cancer Brain Metastasis.
Author Response
- The authors describe some interesting and unusual phenomenon: decrease HSPB2 mRNA in malignant breast tissue comparing to normal tissues, and, at the same time, increase in this mRNA in patients with relapse and poor survival. So far, in most cases levels of chaperones (e.g., Hsp70, HSPB1) were found to increase both during transformation and cancer progression. Thus, if HSPB2 plays anti-apoptotic role as the authors mention, it is completely unclear why its levels drops in cancer where anti-apoptotic protection is indispensable feature and one may expect rather increase than decrease in its levels. The major concern of this work is that the authors studied only expression of HSPB2 mRNA rather than protein which was done in most previous studies, and it is well-known that protein expression (which is more relevant) may not correlate with mRNA expression. Therefore, it would be much more convincing if the authors find some data in databases or elsewhere regarding correlation between expression of HSPB2 mRNA and protein.
We thank the Reviewer for this constructive comment. Indeed, as shown by transcriptomics and proteomics data of HSPB2 expression in various breast cancer cell lines, deposited to the Expression Atlas database:
(https://www.ebi.ac.uk/gxa/search?geneQuery=%5B%7B%22value%22%3A%22HSPB2%22%2C%22category%22%3A%22symbol%22%7D%5D&species=homo%20sapiens&conditionQuery=%5B%7B%22value%22%3A%22breast%20cancer%22%7D%5D&bs=%7B%22homo%20sapiens%22%3A%5B%22CELL_LINE%22%5D%7D&ds=%7B%22kingdom%22%3A%5B%22animals%22%5D%7D#baseline),
the mRNA levels of HSPB2 are not depicted in HSPB2 protein levels. This is also confirmed by the Human Protein Atlas database, where HSPB2 mRNA expression in various tissue is not correlated with HSPB2 protein expression, as well (https://www.proteinatlas.org/ENSG00000170276-HSPB2/tissue).
Prompted by the Reviewer’s comment and these data, we discuss this in the revised manuscript:
Page 11 (Lines 345 – 363): Despite the fact that many sHSPs are overexpressed in a wide range of solid tumors [48], we observed that HSPB2 gene transcription is predominantly downregulated in malignant breast tumors, compared to their adjacent non-cancerous breast tissues. In line with our findings, a recent comprehensive transcriptomic study of the HSP gene family in BrCa patients from both The Cancer Genome Atlas (TCGA) and the Molecular Taxonomy of Breast Cancer International Consortium (METABRIC) cohorts revealed that HSPB2 gene expression levels are profoundly downregulated in all BrCa molecular subtypes as compared to normal breast tissues [49]. Salhia et al. have also demonstrated underexpression of HSPB2 in BrCa samples, due to HSPB2 gene deletion [50]. Other studies support the downregulation of HSPB2 in cancer, as well; specifically, HSPB2 mRNA was lower in esophageal squamous cell carcinoma cell lines, due to hypermethylation of the promoter of HSPB2 gene [51], while HSPB2 mRNA was also found to be barely expressed in pancreatic cancer [52]. Since all the aforementioned studies note the lower mRNA expression of HSPB2, whereas most studies stating the overexpression of HSPs have examined their protein levels, a possible explanation is that HSPB2 protein expression levels does not perfectly correlate with HSPB2 mRNA levels. Indeed, as shown by transcriptomics and proteomics data of HSPB2 expression in various BrCa cell lines, deposited to the Expression Atlas database [53], the lower mRNA levels are not reflected by lower protein expression levels of this molecule.
We also added the appropriate references:
- Salhia, B.; Kiefer, J.; Ross, J.T.; Metapally, R.; Martinez, R.A.; Johnson, K.N.; DiPerna, D.M.; Paquette, K.M.; Jung, S.; Nasser, S., et al. Integrated genomic and epigenomic analysis of breast cancer brain metastasis. PLoS One 2014, 9, e85448, doi:10.1371/journal.pone.0085448.
- Chang, X.; Yamashita, K.; Sidransky, D.; Kim, M.S. Promoter methylation of heat shock protein B2 in human esophageal squamous cell carcinoma. Int J Oncol 2011, 38, 1129-1135, doi:10.3892/ijo.2011.918.
- Papatheodorou, I.; Fonseca, N.A.; Keays, M.; Tang, Y A.; Barrera, E.; Bazant, W.; Burke, M.; Füllgrabe, A.; Fuentes, A.M.-P.; George, N., et al. Expression Atlas: gene and protein expression across multiple studies and organisms. Nucleic Acids Research 2018, 46, D246-D251, doi:10.1093/nar/gkx1158.
- Another point is that no description of how cutoff value cutoff between 2 groups (with high and low HSPB2) was set (Page 3, line 146)
We would like to kindly respond that the classification was done based on the median value of HSPB2 mRNA expression (656.5 RQU). More specifically, patients with HSPB2 mRNA expression value higher than 656.5 RQU were characterized as HSPB2-positive, whereas those with HSPB2 mRNA expression value lower than 656.5 RQU were characterized as HSPB2-negative.
We have stated this in the revised manuscript:
Page 4 (Lines 152-156): […]; this categorizarion was done based on the median value of HSPB2 mRNA expression (656.5 RQU). More specifically, patients with HSPB2 mRNA expression value higher than 656.5 RQU were characterized as HSPB2-positive, whereas those with HSPB2 mRNA expression value lower than 656.5 RQU were characterized as HSPB2-negative.
- Also, at least 2 refs are absent:
- Chang et al. Promoter methylation of heat shock protein B2 in human esophageal squamous cell carcinoma. Int J Oncol 38: 1129 (2011), where decrease in HspB2 mRNA was found in malignant tissue.
- Salhia et al. Integrated Genomic and Epigenomic Analysis of Breast Cancer Brain Metastasis. PLOS Volume 9 | Issue 1 | e85448 (2014), where underexpression/deletion of HspB2 gene was observed in Breast Cancer Brain Metastasis.
As suggested by the Reviewer, we have added the following references in the revised manuscript:
- Salhia, B.; Kiefer, J.; Ross, J.T.; Metapally, R.; Martinez, R.A.; Johnson, K.N.; DiPerna, D.M.; Paquette, K.M.; Jung, S.; Nasser, S., et al. Integrated genomic and epigenomic analysis of breast cancer brain metastasis. PLoS One 2014, 9, e85448, doi:10.1371/journal.pone.0085448.
- Chang, X.; Yamashita, K.; Sidransky, D.; Kim, M.S. Promoter methylation of heat shock protein B2 in human esophageal squamous cell carcinoma. Int J Oncol 2011, 38, 1129-1135, doi:10.3892/ijo.2011.918.
The Authors wish to thank the Reviewer for the constructive comments that led to the improvement of the current manuscript.

Reviewer 2 Report
Authors: Aimilia D. Sklirou, Despoina D. Gianniou, Paraskevi Karousi, Christina Cheimonidi, Georgia Papachristopoulou, Christos K. Kontos, Andreas Scorilas, Ioannis P. Trougakos
Title: High mRNA expression levels of heat shock protein family B member 2 (HSPB2) are associated with breast cancer patients’ relapse and poor survival
REVIEWER'S COMMENTS:
The Authors have performed an interesting study and submitted intriguing data that may have a prognostic significance for breast cancer. The submitted manuscript is well written and designed. However, there are serious questions toward the methodology used:
1. Please explain why you have chosen genes GAPDH and HMBS as the reference genes. Whether you examined the expression stability of these genes in terms of your experiments? If yes, by which method? And whether you considered any other candidates on roles of reference genes in the present study?
2. According to data of Primer-BLAST service, primers to HMBS can amplify, besides the target fragment of 150 nucleotides, a number of concomitant (non-target) products including as minimum 7 variants of amplicons shorter than 1000 nucleotides. Therefore, you had to examine by electrophoresis whether this pair of the primers did amplify the specific product (i.e. HMBS gene fragment in 150 nucleotides) only. Did you do it by means of electrophoresis?
3. Please explain why you decided to use MCF-7 cells as a calibrator in your comparative study? Why not other breast cancer cell lines? MCF-7 cell line is a model of only one (luminal A) subtype, while you had the tumor samples of 4 different breast cancer subtypes.
4. Why you do not confirm your PCR data by the data on HSPB2 protein expression obtained with Western blotting? (The Western blot results might nicely support your conclusions and discussion.)
Minor criticism:
It seems strange that you have inserted in Discussion a paragraph about HSP90 and its inhibition, while your paper is dedicated to HSPB2. Instead, you could discuss the cancer-promoting roles of HSPB in more details with mentioning its implication in EMT, cancer stemness, metastasis spread etc.
Author Response
- Please explain why you have chosen genes GAPDH and HMBS as the reference genes. Whether you examined the expression stability of these genes in terms of your experiments? If yes, by which method? And whether you considered any other candidates on roles of reference genes in the present study?
We would like to kindly respond that we examined the expression stability of four (4) candidate reference genes encoding glyceraldehyde 3-phosphate dehydrogenase (GAPDH), hydroxymethylbilane synthase (HMBS), hypoxanthine phosphoribosyltransferase 1 (HPRT1), and beta-2 microglobulin (B2M) in eight (8) breast cancer samples. The expression levels of all the candidate reference genes were shown to have similar co-variation among the cDNAs. In order to choose the most appropriate reference genes, we also searched for respective information in the current literature. Thus, HPRT1 was excluded, since studies have shown its upregulation in breast cancer (J. Sedano et al., Cancers. 2020; 12(6):1522. https://doi.org/10.3390/cancers12061522). Among GAPDH, HMBS, and B2M, we chose GAPDH and HMBS, since according to a study by de Kok et al. (Lab Invest 85, 154–159 (2005). https://doi.org/10.1038/labinvest.3700208), these two genes are more suitable reference genes when working with breast cancer samples (based on the calculated coefficients of correlation, which were 0.93 for HMBS, 0.82 for GAPDH and 0.53 for B2M).
We briefly explained this in the revised manuscript:
Page 3 (Lines 126-130): […] moreover, specific primer pairs were designed for housekeeping genes encoding glyceraldehyde-3-phosphate dehydrogenase (GAPDH) and hydroxymethylbilane synthase (HMBS), which were chosen among others as reference genes, according to supporting literature data [27] and our preliminary results in a random sample of our cohort of BrCa specimens.
We also added the respective reference:
- de Kok, J.B.; Roelofs, R.W.; Giesendorf, B.A.; Pennings, J.L.; Waas, E.T.; Feuth, T.; Swinkels, D.W.; Span, P.N. Normalization of gene expression measurements in tumor tissues: comparison of 13 endogenous control genes. Lab Invest 2005, 85, 154-159, doi:10.1038/labinvest.3700208.
- According to data of Primer-BLAST service, primers to HMBS can amplify, besides the target fragment of 150 nucleotides, a number of concomitant (non-target) products including as minimum 7 variants of amplicons shorter than 1000 nucleotides. Therefore, you had to examine by electrophoresis whether this pair of the primers did amplify the specific product (i.e. HMBS gene fragment in 150 nucleotides) only. Did you do it by means of electrophoresis?
We thank the Reviewer for this remark. We have checked the specificity of all the primer pairs used in qPCR assays. Indeed, the primer pair used to amplify HMBS could possibly amplify some off-targets too; however, a perfect match of the primers is only achieved on the HMBS mRNA sequence. Moreover, we optimized the qPCR assays by standardizing the primer concentration and the thermal protocol, so as to observe a unique melting curve for each assay, equivalent to a unique amplicon for each qPCR assay. Last, we used agarose gel electrophoresis, to ensure that the length of each PCR product corresponds to the desired one.
We have added this information in the revised manuscript:
Page 3 (Lines 138-140): We optimized the qPCR assays by standardizing the primer concentration and the thermal protocol, so as to observe a unique melting curve for each assay. The uniqueness and specificity of each amplicon were also assessed by agarose gel electrophoresis.
- Please explain why you decided to use MCF-7 cells as a calibrator in your comparative study? Why not other breast cancer cell lines? MCF-7 cell line is a model of only one (luminal A) subtype, while you had the tumor samples of 4 different breast cancer subtypes.
As defined by Livak and Schmittgen (Methods, Volume 25, Issue 4, 2001, https://doi.org/10.1006/meth.2001.1262), when performing relative quantification, the calibrator is used to render minor corrections of qPCR efficiency differences, occurring among distinct qPCR runs, feasible. Therefore, we could use a cDNA deriving from whichever breast cancer cell line, without affecting the relative quantification results.
We provided a short explanation in the revised manuscript:
Page 3 (Lines 140-142): The cDNA derived from the MCF-7 cells was used as a calibrator to render “-∆Ct” values calculated in distinct qPCR runs comparable.
- Why you do not confirm your PCR data by the data on HSPB2 protein expression obtained with Western blotting? (The Western blot results might nicely support your conclusions and discussion.)
We thank the Reviewer for this constructive comment. The reason for which we have not performed a Western blot assay is that HSPB2 mRNA and protein levels do not seem to correlate. As shown by transcriptomics and proteomics data of HSPB2 expression in various breast cancer cell lines, deposited to the Expression Atlas database:
(https://www.ebi.ac.uk/gxa/search?geneQuery=%5B%7B%22value%22%3A%22HSPB2%22%2C%22category%22%3A%22symbol%22%7D%5D&species=homo%20sapiens&conditionQuery=%5B%7B%22value%22%3A%22breast%20cancer%22%7D%5D&bs=%7B%22homo%20sapiens%22%3A%5B%22CELL_LINE%22%5D%7D&ds=%7B%22kingdom%22%3A%5B%22animals%22%5D%7D#baseline),
the mRNA levels of HSPB2 are not depicted in the HSPB2 protein levels. This is also confirmed by the Human Protein Atlas database, where HSPB2 mRNA expression in various tissue is not correlated with HSPB2 protein expression, as well (https://www.proteinatlas.org/ENSG00000170276-HSPB2/tissue).
In response to the Reviewer’s comment, we added the following text in the Discussion section:
Page 11 (Lines 345 – 363): Despite the fact that many sHSPs are overexpressed in a wide range of solid tumors [48], we observed that HSPB2 gene transcription is predominantly downregulated in malignant breast tumors, compared to their adjacent non-cancerous breast tissues. In line with our findings, a recent comprehensive transcriptomic study of the HSP gene family in BrCa patients from both The Cancer Genome Atlas (TCGA) and the Molecular Taxonomy of Breast Cancer International Consortium (METABRIC) cohorts revealed that HSPB2 gene expression levels are profoundly downregulated in all BrCa molecular subtypes as compared to normal breast tissues [49]. Salhia et al. have also demonstrated underexpression of HSPB2 in BrCa samples, due to HSPB2 gene deletion [50]. Other studies support the downregulation of HSPB2 in cancer, as well; specifically, HSPB2 mRNA was lower in esophageal squamous cell carcinoma cell lines, due to hypermethylation of the promoter of HSPB2 gene [51], while HSPB2 mRNA was also found to be barely expressed in pancreatic cancer [52]. Since all the aforementioned studies note the lower mRNA expression of HSPB2, whereas most studies stating the overexpression of HSPs have examined their protein levels, a possible explanation is that HSPB2 protein expression levels does not perfectly correlate with HSPB2 mRNA levels. Indeed, as shown by transcriptomics and proteomics data of HSPB2 expression in various BrCa cell lines, deposited to the Expression Atlas database [53], the lower mRNA levels are not reflected by lower protein expression levels of this molecule.
We also added the appropriate references:
- Salhia, B.; Kiefer, J.; Ross, J.T.; Metapally, R.; Martinez, R.A.; Johnson, K.N.; DiPerna, D.M.; Paquette, K.M.; Jung, S.; Nasser, S., et al. Integrated genomic and epigenomic analysis of breast cancer brain metastasis. PLoS One 2014, 9, e85448, doi:10.1371/journal.pone.0085448.
- Chang, X.; Yamashita, K.; Sidransky, D.; Kim, M.S. Promoter methylation of heat shock protein B2 in human esophageal squamous cell carcinoma. Int J Oncol 2011, 38, 1129-1135, doi:10.3892/ijo.2011.918.
- Papatheodorou, I.; Fonseca, N.A.; Keays, M.; Tang, Y A.; Barrera, E.; Bazant, W.; Burke, M.; Füllgrabe, A.; Fuentes, A.M.-P.; George, N., et al. Expression Atlas: gene and protein expression across multiple studies and organisms. Nucleic Acids Research 2018, 46, D246-D251, doi:10.1093/nar/gkx1158.
- It seems strange that you have inserted in Discussion a paragraph about HSP90 and its inhibition, while your paper is dedicated to HSPB2. Instead, you could discuss the cancer-promoting roles of HSPB in more details with mentioning its implication in EMT, cancer stemness, metastasis spread etc.
The role of HSPB2 in cancer has only been poorly explored, compared to the role of other HSPs, such as HSP70 and HSP90. This is the reason we also refer to HSP90 in the Discussion section. We mention this in our revised manuscript, and also refer to the role of HSPB2 in pancreatic cancer:
Page 12 (Lines 381-385): Furthermore, although the role of HSPB2 has not been investigated so widely as the role of other HSPs, it was found to confer resistance to apoptosis in human BrCa cell lines, as it inhibited the apical caspase activation in the extrinsic apoptotic pathway [26], as well as to inhibit pancreatic cancer cell proliferation via activating targets of the TP53 signaling pathway, such as the RPRM, ADGRB1, and STEAP3 genes [52]. In addition, it has recently been hypothesized that inhibition of HSPB2 by miR-17-5p promotes cell proliferation, migration, and invasion of colon cancer cell lines [61].
We also added the following reference:
- Yu, W.; Wang, J.; Li, C.; Xuan, M.; Han, S.; Zhang, Y.; Liu, P.; Zhao, Z. miR-17-5p promotes the invasion and migration of colorectal cancer by regulating HSPB2. J Cancer 2022, 13, 918-931, doi:10.7150/jca.65614.
The Authors wish to thank the Reviewer for the constructive comments that led to the improvement of the current manuscript.

Reviewer 3 Report
In the paper entitled “High mRNA expression levels of heat shock protein family B 2 member (HSPB2) are associated with breast cancer patients’ relapse and poor survival” by Sklirou and colleagues, the authors try to find novel non-invasive biomarkers that could help clinicians to estimate patience recurrence and survival.
Despite the limitations of the study represented by reduced cohort size, low number of non-cancerous tissue, and not equivalent stratification of patients, its results are promising.
The authors have demonstrated that HSPB2 mRNA is downregulated in malignant breast tumors compared to non-cancerous tissues. Moreover, increased HSPB2 mRNA expression is a biomarker for poor DFS and OS, independently by clinicopathological factors. In a such way HSPB2 expression could be used to asses clinical prognosis of patients.
Author Response
In the paper entitled “High mRNA expression levels of heat shock protein family B 2 member (HSPB2) are associated with breast cancer patients’ relapse and poor survival” by Sklirou and colleagues, the authors try to find novel non-invasive biomarkers that could help clinicians to estimate patience recurrence and survival.
Despite the limitations of the study represented by reduced cohort size, low number of non-cancerous tissue, and not equivalent stratification of patients, its results are promising.
The authors have demonstrated that HSPB2 mRNA is downregulated in malignant breast tumors compared to non-cancerous tissues. Moreover, increased HSPB2 mRNA expression is a biomarker for poor DFS and OS, independently by clinicopathological factors. In a such way HSPB2 expression could be used to asses clinical prognosis of patients.
We thank the Reviewer for the positive appraisal of our original research article. We are aware of the limitations of our work and have discussed them in the revised version of our manuscript:
Page 12 (lines 394-399): Nevertheless, our study is characterized by some limitations that need to be addressed. Firstly, our cohort size is of medium size, and the number of non-cancerous breast tissue specimens is rather small. In addition, the patients’ cohort was not equivalently stratified in the defined subgroups, which could diminish the obtained findings. Future studies should be conducted to further evaluate the role of HSPB2 in BrCa prognosis.
The Authors wish to thank the Reviewer for the constructive comments that led to the improvement of the current manuscript.

Round 2
Reviewer 1 Report
No comments
Reviewer 2 Report
Dear Authors,
After the revisions you performed, I am pleased to recommend to accept your manuscript for publications in IJMS.
Good luck in your research work!